# EYE-503: A Novel Retinoic Acid Drug for Treating Retinal Neurodegeneration

**DOI:** 10.3390/ph16071033

**Published:** 2023-07-20

**Authors:** Sha Liu, Yuke Ji, Huan Li, Ling Ren, Junya Zhu, Tianjing Yang, Xiumiao Li, Jin Yao, Xin Cao, Biao Yan

**Affiliations:** 1The Affiliated Eye Hospital, Nanjing Medical University, Nanjing 210093, China; liusha_2020@126.com (S.L.); jyuke202212@126.com (Y.J.); lhuan202303@163.com (H.L.); zhujunya2307@163.com (J.Z.); ytj_2018@126.com (T.Y.); xiumiaol11@163.com (X.L.); 2The Fourth School of Clinical Medicine, Nanjing Medical University, Nanjing 210093, China; 3Eye Institute and Department of Ophthalmology, Eye and ENT Hospital, State Key Laboratory of Medical Neurobiology, Fudan University, Shanghai 200433, China; 22111260014@m.fudan.edu.cn; 4Institute of Clinical Science, Zhongshan Hospital, Fudan University, Shanghai 200433, China; 5NHC Key Laboratory of Myopia, Key Laboratory of Myopia, Chinese Academy of Medical Sciences, Shanghai 200031, China; 6Shanghai Key Laboratory of Visual Impairment and Restoration, Fudan University, Shanghai 200433, China

**Keywords:** retinoic acid, retinal neurodegeneration, retinal ganglion cell, optic nerve crush, neuroprotective effects

## Abstract

Retinal neurodegeneration is a major cause of vision loss. Retinoic acid signaling is critical for the maintenance of retinal function, and its dysfunction can cause retinal neurodegeneration. However, the therapeutic effects of retinoic acid drugs on retinal neurodegeneration remain unclear. In this study, we designed a novel retinoic acid drug called EYE-503 and investigated its therapeutic effects of EYE-503 on retinal neurodegeneration. The optic nerve crush (ONC) model was selected for the retinal neurodegeneration study. H&E staining, TUNEL staining, immunofluorescence staining, and visual electrophysiology assays were performed to determine the role of EYE-503 in retinal neurodegeneration in vivo. The CCK-8 assay, EdU incorporation assay, PI staining, and flow cytometry assays were performed to investigate the effects of EYE-503 administration on retinal neurodegeneration in vitro. The potential mechanism of EYE-503 in retinal neurodegeneration was investigated by network pharmacology and Western blots. The results showed that EYE-503 administration had no detectable cytotoxicity and tissue toxicity. EYE-503 administration alleviated ONC-induced retinal injury and optic nerve injury in vivo. EYE-503 administration attenuated retinal ganglion cell apoptosis, inhibited reactive gliosis, and retarded the progression of retinal neurodegeneration. Mechanistically, EYE-503 regulated retinal neurodegeneration by targeting the JNK/p38 signaling pathway. This study suggests that EYE-503 is a promising therapeutic agent for retinal neurodegenerative diseases.

## 1. Introduction

Retinal neurodegeneration is a leading cause of irreversible blindness in several ocular diseases, such as diabetic retinopathy, glaucoma, and optic neuropathy [1,2]. It is often characterized by the apoptosis of retinal ganglion cells (RGCs), axonal degeneration, and reactive gliosis [3]. Previous studies have shown that retinal neurodegeneration is caused by several pathological factors, such as the disruption of neurotrophic factors, mitochondrial dysfunction, inflammation, glutamate excitatory toxicity, calcium overload, and oxidative stress [4,5,6,7], which in turn leads to axon degeneration, RGC death, and eventual vision loss. Currently, several agents have been used to alleviate RGC injury and axonal degeneration to improve visual functions, such as neurotrophic factors, glutamate receptor antagonists, calcium channel blockers, and antioxidants [8,9,10,11,12]. However, the therapeutic effects are still far from satisfactory. Thus, searching for new agents for treating retinal neurodegeneration is still required.

Retinoic acid (RA) is an active metabolite of vitamin A that regulates many biological processes, including cell differentiation, proliferation, apoptosis, and embryonic development [13,14,15]. Due to their powerful and broad-spectrum activities, RA and RA-mediated signaling have been implicated in the pathogenesis of several human diseases, particularly in neurodegenerative diseases, including Alzheimer’s disease, Parkinson’s disease, ischemic stroke, and amyotrophic lateral sclerosis [16,17,18,19]. Previous studies have revealed that RA exerts neuroprotective effects via increasing neuronal activity, attenuating neural inflammation, suppressing microglial activation, improving neural stem cell function, and maintaining blood–brain barrier integrity [20,21,22]. In addition, RA has been identified as a crucial regulator of the integrity of corneal epithelium, dendritic growth of photoreceptors, and the pigmentation of retinal pigment epithelium [23,24]. Based on the aforementioned evidence, we speculate that RA is a promising drug for treating retinal neurodegeneration. However, its poor water solubility and drug resistance limit the application of current RA drugs in clinical issues [25,26].

In this study, we designed a novel RA drug, EYE-503, for the treatment of retinal neurodegeneration. EYE-503 was designed based on tazarotene through the substitution of benzoic acid with aromatic rings and using a racemic sulfoxide derivative bearing a 5-pyrimidine-acid skeleton to improve water solubility. Then, the neuroprotective effects of EYE-503 on retinal degeneration were evaluated. The results showed that EYE-503 administration could ameliorate RGC degeneration and axonal degeneration and reduce retinal reactive gliosis following optic nerve injury via the JNK/p38 MAPK signaling pathway. This study provides a novel drug for the treatment of retinal neurodegeneration, which will provide a novel method to improve visual function in patients with retinal neurodegeneration.

## 2. Results

### 2.1. EYE-503 Has No Detectable Cytotoxicity and Tissue Toxicity

EYE-503 (10 μM) or 0.9% saline was administered into mouse eyes via intraocular injection. On day 7, following intraocular injection, retinal histological changes were observed by H&E staining (Figure 1a). As shown in Figure 1a, EYE-503 administration did not cause detectable pathological changes in histological structures. Similar results were observed in TUNEL staining experiments. EYE-503 administration did not induce a marked increase in retinal apoptosis (Figure 1b). CCK-8 assays were performed to detect whether EYE-503 had cytotoxicity in vitro. The results showed that EYE-503 did not have toxic effects on RGCs and Müller cells at the tested concentrations ranging from 100 nM to 10 μM (Figure 1c,d). Annexin V-FITC/PI double staining assays showed that EYE-503 did not cause detectable apoptosis at the experimental concentrations (500 nM and 1 μM) (Figure 1e,f). Collectively, these results suggest that EYE-503 has no obvious cytotoxicity or tissue toxicity.

### 2.2. EYE-503 Administration Inhibits Retinal Reactive Gliosis and Contributes to RGC Survival

We established an ONC model using C57BL/6J mice to investigate the role of EYE-503 in retinal neurodegeneration. EYE-503 (10 μM) or 0.9% saline was injected into the vitreous immediately after surgery. In retinal whole-mounts and RGC counting experiments, the densities of RGCs in the Ctrl group, ONC group, 0.9% saline-treated group (ONC + 0.9% saline), and EYE503-treated group (ONC + EYE-503) were 2005 ± 149/mm^2^, 978 ± 89/mm^2^ (48.8% survival), 1079 ± 85/mm^2^ (53.8% survival), and 1713 ± 183/mm^2^ (85.4% survival), respectively. There were no significant differences in RGC densities between the ONC group and 0.9% saline-treated group plus ONC group. Compared with the 0.9% saline-treated group, EYE-503 administration led to a marked increase in RGC survival by approximately 31.6% (*p* < 0.05) (Figure 2a). TUBB3 staining and NeuN staining further verified the beneficial effects of EYE-503 on RGC survival (Figure 2b,c). Previous studies revealed that reactive gliosis is tightly associated with increased expression of GS and GFAP [27,28,29]. Compared with 0.9% saline-treated retina, EYE-503 administration significantly reduced retinal reactive gliosis, as shown by decreased glutamine synthetase (GS) and glial fibrillary acidic protein (GFAP) staining (Figure 2d,e). Taken together, these results suggest that EYE-503 administration contributes to RGC survival and inhibits retinal reactive gliosis in vivo.

### 2.3. EYE-503 Administration Alleviates ONC-Induced Retinal Injury and Optic Nerve Injury In Vivo

H&E staining was conducted to detect the change in the number of retinal ganglion cells in the GCL layer. Compared with the control group, ONC treatment led to a marked decrease in the number of retinal ganglion cells, while EYE-503 administration partially alleviated ONC-induced RGC loss (Figure 3a). Histological observations revealed that the myelin sheath was swollen and fragmented 7 days after optic nerve insult; this damage to the optic nerve myelin sheath was alleviated by EYE-503 administration (Figure 3b). These results suggest that EYE-503 exerts protective effects on ONC-induced retinal injury.

We also determined the effects of EYE-503 administration on retinal function. Flash visual-evoked potentials (F-VEPs) were recorded on day 7 following ONC induction. F-VEP changes in the mice across different groups are shown in Figure 3c. The latency of the P1 wave represented the time of signal transduction from the retina to the occipital cortex. On day 7 after ONC injury, the latency of P1 (87.3 ± 4.09 ms) markedly increased, and the amplitude of N1-P1 (4.6 ± 1.59 μV) significantly decreased in ONC mice compared with the control group (47.5 ± 4.64 ms, 17.3 ± 0.96 μV). No significant difference in F-VEP N1-P1 amplitude and P1 latency was detected between the ONC group and ONC + 0.9% saline-treated group. As expected, intravitreal injection of EYE-503 led to a marked increase in the amplitude of N1-P1-wave at day 7 following ONC injury compared with 0.9% saline treatment (*p* < 0.05). By contrast, P1-wave latency obviously decreased from (87.3 ± 4.09 ms) to (71.7 ± 4.84 ms) in the EYE-503-treated group (Figure 3d,e). These data suggest that EYE-503 administration could ameliorate ONC-induced retinal visual function.

### 2.4. EYE-503 Administration Regulates Müller Cell and RGC Function In Vitro

Based on the above-mentioned results, we speculated that EYE-503 exerted neuroprotective effects during retinal neurodegeneration by regulating RGC and Müller cell function. Primary mouse RGCs were cultured and identified with TUBB3 staining [30,31] (>85%, Figure 4a). The primary mouse Müller cells were identified with GS and GFAP staining (>90%, Figure 5a). CCK-8 assays showed that H_2_O_2_ treatment significantly decreased the viability of RGCs and Müller cells, whereas EYE-503 administration alleviated oxidative stress-induced injury of RGCs and Müller cells compared with the H_2_O_2_ treatment group (Figure 4b and Figure 5b). EdU immunofluorescence staining indicated that EYE-503 significantly increased the proliferative ability of Müller cells under oxidative stress (Figure 5c,d). Then, we detected whether EYE-503 regulated H_2_O_2_-induced apoptosis using PI staining and Annexin V-FITC/PI double staining assays. The results demonstrated that EYE-503 administration significantly reduced the harmful effects of oxidative stress on RGCs and Müller cells (Figure 4c–f and Figure 5f,h–j). For Müller cells, we further performed rhodamine 123 staining to evaluate the effects of EYE-503 treatment on H_2_O_2_-induced mitochondrial potentials. Rhodamine 123 staining showed that H_2_O_2_-induced reduction in mitochondrial potential was partially reversed by EYE-503 administration in Müller cells (Figure 5e,g). Collectively, these results suggest that EYE-503 administration can protect against oxidative stress-induced apoptosis in RGCs and Müller cells in vitro.

### 2.5. EYE-503 Administration Induces Bcl-2 Downregulation and Bax Translocation

B-cell CLL/Lymphoma 2 (Bcl-2) and Bcl-2 associated x (Bax) protein are homogenous proteins that have opposite effects on cell function [32]. Western blot assays revealed that the levels of Bcl-2 or Bax expression in RGCs were decreased or increased following H_2_O_2_ treatment, respectively. EYE-503 administration inhibited RGC apoptosis, as shown by decreased expression of Bax and increased expression of Bcl-2 (Figure 6a,b). These data suggest that EYE-503 administration can suppress the activation of apoptotic signaling and contributes to the survival of retinal cells.

### 2.6. The Targets of EYE-503 Were Predicted by Network Pharmacology

The chemical structure of EYE-503 is shown in Figure 7a. The SwissTargetPrediction and Superpred databases were used to search for the potential targets of EYE-503. After filtering out these duplicate targets, 111 potential targets of EYE-503 were retrieved. In total, 1115 targets of retina neurodegeneration were obtained from the DrugBank, OMIM, and GeneCard databases. After mapping EYE-503-related targets and disease-related targets in a Venn diagram, 38 overlapped targets were accessed (Figure 7b). To identify the core protein interaction of EYE-503 intervention for retinal neurodegeneration, 38 drug-disease common targets were inducted into the STRING database. The target network was made up of 38 nodes and 109 edges, with an average node degree of 5.74. The PPI network diagram of the EYE-503 against retinal neurodegeneration was constructed by Cytoscape 3.0.1 (Figure 7c).

Next, the metascape database was used for GO and KEGG pathway enrichment analysis to investigate the potential signaling pathways or biological processes based on the 38 drug-disease common targets. The top eight entries for biological process (BP), cellular component (CC), and molecular function (MF) were screened by GO enrichment analysis (Figure 7d). The BP of drug-disease targets was highly enriched in the modulation of synaptic signaling, sensory perception of pain, cognition, and regulation of neuron death. The most highly enriched CC were membrane raft, membrane microdomain, external side of the plasma membrane, synaptic membrane, neuronal cell body, etc. MF was mainly enriched in protein serine/threonine/tyrosine kinase activity, serotonin binding, amine binding, neurotransmitter receptors, etc. The 10 most important KEGG signaling pathways related to retinal neurodegeneration with the condition of *p* < 0.05 are shown in Figure 7e. The calcium signaling pathway, neuroactive ligand-receptor interaction, Rap1 signaling pathway, cancer pathway, MAPK signaling pathway, and neurotrophin signaling pathway were included. EYE-503 may act on these pathways against retinal neurodegeneration.

We speculated that EYE-503 might act on the MAPK signaling pathway against retinal neurodegeneration by network pharmacology. The JNK and p38 branches of the mitogen-activated protein (MAP) kinase pathway can mediate several biological processes, such as proliferation, differentiation, senescence, and apoptosis [33]. It has been reported that aberrant JNK or p38 signaling in neural cells may lead to the occurrence of retinal neurodegenerative diseases [34,35,36]. As shown in Figure 7f,g, stress stimulation led to an obvious increase in the phosphorylated levels of p38 and JNK in RGCs. We found that EYE-503 administration decreased the levels of phospho-JNK and phospho-p38 to various degrees. Collectively, we speculate that EYE-503 administration alleviates retinal cell apoptosis by suppressing JNK/p38 signaling.

## 3. Discussion

RA, as a ligand of the retinoic acid receptor, is involved in retinoid signaling in the nervous system [37]. Previous studies have shown that RA is capable of partially alleviating neuropathy that occurs in animal models of brain diseases [20,38]. The brain and the eye usually share several common characteristics, including similar microvasculature and an underlying gene regulatory mechanism. Thus, we speculate that RA-mediated signaling is tightly associated with the pathogenesis of ocular neurodegeneration.

A variety of pathological factors are involved in the pathogenesis of retinal neurodegeneration, including neuroinflammation, oxidative stress, and mitochondrial dysfunction [2]. Retinoic acid signaling plays an important role in neuronal differentiation and maintenance of neuronal growth [39]. Retinoic acid signaling is activated following optic nerve injury in frogs and fish, which can recover visual function [40]. RA is capable of partially reverting morphologic and functional peripheral neuropathy through its effects on the local induction of nerve growth factor [41]. RA signaling activation contributes to long-term survival following RGC injury [42]. However, the application of RA is limited due to its UV photosensitivity and non-targeted side effects [43]. In this study, we reveal the role of EYE-503 in mechanical stress-induced retinal injury. Morphologic observation shows that intraocular injection of EYE-503 did not cause tissue toxicity and reduce the number of RGCs or destroy RGC function compared with the 0.9% saline-treated group under the physiological condition. Moreover, EYE-503 did not cause cellular toxicity in vitro. Under the pathological condition, EYE-503 administration could reduce RGC injury and alleviate retinal neurodegeneration, suggesting that EYE-503 exerts its neuroprotective effects on retinal function.

Retinal neurodegeneration is a pathological process associated with the involvement of multiple cells and factors. Müller cells, the predominant glial cells of the retina, communicate with retinal neurons via secreted mediators to regulate their survival [44]. Reactive gliosis and subsequent retinal ganglion cell loss are thought to be the final event in the retina after mechanical stress, which in turn leads to vision impairment [45]. Thus, reducing RGC injury is a possible treatment modality for optic nerve injury. We show that EYE-503 treatment significantly reduced ONC-induced RGC degeneration, showing a neuroprotective role during retinal neurodegeneration. The intermediate filament protein, GFAP, is considered an early marker of retinal injury and is commonly used as an index of gliosis hypertrophy. Cellular hypertrophy and increased expression of GFAP are known as the key morphological alterations during retinal neurodegeneration [46,47]. In this study, immunofluorescence staining demonstrates that EYE-503 treatment significantly reduced retinal reactive gliosis, as shown by decreased expression of GFAP and GS. In vitro experiments further reveal that EYE-503 administration affected glial cell viability and reduced proliferative capacity. Moreover, EYE-503 administration could reduce the apoptosis of retinal ganglion cells following ONC injury. Together, these results suggest that EYE-503 plays a neuroprotective role in retinal neurodegeneration.

RGC loss and reactive gliosis are known as the important pathological features of retinal neurodegeneration. Progressive loss of RGCs and their axons is shown as the classic hallmark of retinal neurodegeneration [48]. Müller cells are the major types of retinal glial cells responsible for maintaining retinal homeostasis and ensuring the proper function of a healthy retina. At the early stage of neurodegeneration, activated glial cells play a neuroprotective role in the retina by increasing the expression of cytoprotective factors or restoring neurotransmitter and ion balance. At the late stage of neurodegeneration, proliferative gliosis can accelerate retinal neurodegeneration, causing direct and indirect damage to the neurons and vasculature [49]. This evidence suggests that reactive Müller cells may exert cytoprotective and cytotoxic effects on retinal neurons. Given Müller cells play a dual role in retinal neurodegeneration, we speculated that EYE-503 administration might affect the activation of Müller cells through different signaling pathways at different stages of retinal neurodegeneration. Because RGC loss can be observed throughout the process of retinal neurodegeneration, we mainly investigate the mechanism of EYE-503-mediated RGC protection [50,51].

Previous studies have shown that RA plays an anti-apoptotic role in glutamate receptor activation-induced neuronal injury in the retina. In the middle cerebral artery occlusion (MCAO) mice model, RA treatment attenuated neutrophil accumulation in the infarct lesions by suppressing signal transducers and activators of transcription 1 (STAT1) signaling [20]. RA is also considered a therapeutic target in cerebral ischemia [15]. RA often exerts its molecular actions through RAR and RXR nuclear receptors. EYE-503 is a retinoid compound and a novel RA drug. In our study, we utilized network pharmacology to identify the potential signaling pathways linking EYE-503 and retinal neurodegeneration based on the chemical structure of EYE-503. However, among those signaling pathways, RAR/RXR nuclear receptors were not predicted as the signaling pathway involved in retinal neurodegeneration. By contrast, the MAPK signaling pathway was predicted as a potential pathway involved in RGC degeneration. We thus determined the phosphorylation levels of the key components of JNK/p38 signaling and revealed that EYE-503 treatment significantly reduced the phosphorylated levels of JNK/p38, suggesting that EYE-503 exerts its neuroprotective effects on retinal neurodegeneration through JNK/p38 signaling.

## 4. Materials and Methods

### 4.1. Animals

C57BL/6J mice (6- to 8-week-old male) were purchased from the Animal Core Facility of Nanjing Medical University (Nanjing, China). They were housed under the pathogen-free condition with a temperature of 25 ± 2 °C, humidity of 50 ± 5%, and maintained at a 12 h light/dark cycle with free access to food and water.

### 4.2. Optic Nerve Crush (ONC) Model

The mice were anesthetized with an intraperitoneal injection of ketamine (80 mg/kg) and xylazine (10 mg/kg). A few drops of topical anesthesia on the ocular surface blocked the reflex. The fornix conjunctiva was incised with surgical scissors. The ocular muscle and orbital fat were bluntly separated without destroying the surrounding blood vessels and separated along the posterior eye until the optic nerves were exposed. The optic nerves were crushed with cross-action forceps for 10 sec. After surgery, the fundus was checked fundoscopically to confirm the absence of ischemic damage. Antibiotic ointment was applied to avoid potential infection. In the following experiment, the mice that had retinal ischemia or cataract were excluded.

### 4.3. Intravitreal Injection

Intravitreal injection was performed immediately following optic nerve crush. The mice were anesthetized with ketamine (80 mg/kg) and xylazine (10 mg/kg) and then injected intravitreally through an operating microscope (66 Vision Tech, Suzhou, China). Approximately 2 μL of drug or 0.9% saline were injected using a 33-gauge microneedle. The needle was inserted approximately 1 mm away from the corneal limbus to avoid injuries to the lens. Tobramycin ointment was applied to avoid bacterial infections after intravitreal injection.

### 4.4. Flash Visual Evoked Potentials (F-VEP) Recording

Retinal visual function was detected using full-field flash electroretinography. The mice were anesthetized with a combination of ketamine (80 mg/kg) and xylazine (10 mg/kg). The pupils were dilated using the drop of 1% tropicamide and fixed on the experimental platform. For F-VEP recording, subcutaneous platinum needle electrodes were inserted into the skin in the middle of two ears. The needle-type electrodes in the cheek pouches and at the base of the tail served as the reference and ground electrode. Brief white light flashes were delivered at 1 Hz frequency (bandpass filtered at 1–100 Hz) in background luminance of 3 cd·s/m^2^ with 63 sweeps averaged per recording. The flash duration was <5 ms. VEP responses were measured with the GOTEC visual testing system (GOTEC Medical, Chongqing, China). The latency and amplitude were averaged from 3 trials per eye.

### 4.5. Hematoxylin and Eosin (H&E) Staining

Following optic nerve crush, the retinas were dissected from Fekete’s solution-fixed eyes and fixed in 4% paraformaldehyde (BL539A, Biosharp, Hefei, China) for 24 h. Subsequently, the retinas were embedded in paraffin and sectioned at 5-µm thickness. The de-paraffinized and rehydrated cross-sections were stained with hematoxylin and eosin (H&E, BP-DL001, Sbjbio, Nanjing, China). The serial sections within 100 µm of the optic nerve head were used for the parallel comparisons of different groups. The images were taken under a light microscope (IX73-TH4-200, Olympus, Japan). The number of retinal ganglion cells in GCL layers was calculated. H&E staining was conducted as per the above-mentioned steps. The number of swollen or fragments of the optic nerve was counted.

### 4.6. Terminal Deoxynucleotidyl Transferase dUTP Nick-End Labeling (TUNEL) Assay

Apoptosis was detected using the One Step TUNEL Apoptosis Assay Kit (C1088, Beyotime, Shanghai, China). The paraffin sections were de-paraffinized by xylene and ethanol treatment and then cultivated in proteinase K working solution (20 μg/mL) for 15 min at room temperature. The positive control group was treated with DNase-I reaction solution (C1082) for 15 min. Subsequently, the sections were washed with PBS and incubated with a TUNEL reaction mixture at 37 °C for 1 h. The nuclear morphology of TUNEL-positive cells was stained with 4, 6-diamidino-2-phenylindole (DAPI, 1:1500 dilution, C1002, Biosharp, Hefei, China). The images were captured under a fluorescence microscope (IX73-TH4-200, Olympus, Japan).

### 4.7. Immunohistochemistry

The eyes were enucleated and fixed in Fekete’s solution for 2 h. The retinas were dissected out and fixed in 4% paraformaldehyde overnight at 4 °C. After immersing in 30% sucrose for 48 h, retinal tissues were embedded in the optimum cutting temperature compound mounting media (4583, Sakura, Torrance, CA, USA) and sectioned into 10-μm slices. After blocking in PBS containing 5% bovine serum albumin (BSA) and 1% Triton X-100 at 37 °C for 45 min, the sections were incubated with glutamine synthetase (GS, 1:400 dilution, ab228590, Abcam, Cambridge, UK), Glial fibrillary acidic protein (GFAP, 1:50 dilution, SC-33673, Santa Cruz, CA, USA), NeuN (1:400 dilution, ab177487, Abcam, Cambridge, UK), or TUBB3(1:500 dilution, ab18207, Abcam, Cambridge, UK) antibody overnight at 4 °C. After washing with PBS containing 0.1% Tween 20 (PBST), the sections were incubated with the secondary antibody for 2 h at room temperature in the dark. Finally, retinal sections were counterstained with DAPI for 10 min and observed under a fluorescence microscope.

### 4.8. Retinal Flat-Mounts and TUBB3 Staining

After the optic nerve crush, the eyes were fixed with 4% paraformaldehyde for 30 min. Retinal flat-mount sections were isolated, dissected, and flattened on the slides. Retinal tissues were then permeabilized with 5% BSA and 1% Triton X-100 at 37 °C for 45 min and stained with TUBB3 (1:200 dilution) overnight at 4 °C for 12 h. Alexa Fluor 594 goat anti-rabbit IgG (1:500 dilution, A11012, Invitrogen, Carlsbad, California, USA) was used as the secondary antibody. In total, 40× tile scan images in the peripheral and mid-peripheral regions of each quadrant of the retina were obtained by a fluorescence microscope. Two images of each quadrant were taken at equal distances from the optic papilla. The number of TUBB3 positive cells per mm^2^ was counted in all quadrants by Image J and averaged.

### 4.9. Cell Culture and Treatment

The retinas of postnatal day 0-3 mice were dissected under a microscope and digested by a papain solution (5 mg/mL). After digesting at 37 °C for 30 min, RGCs were purified by incubating with the anti-macrophage antibody (CLAD31240, Cedar Lane, Burlington, ON, Canada). Next, the cell suspensions were transferred to a 100 mm dish coated with anti-Thy1.2 antibody for secondary purification (MCA02R, Bio-Rad, Hercules, CA, USA). The purified RGCs were triturated with a pipette and cultured in the complete medium (CM-M122, Procell, Wuhan, China) [31,52]. The mouse primary Müller cells were obtained from 7 to 10 day postnatal mice. Retinal tissues were isolated and digested for 30 min with 0.25% trypsin, followed by Dulbecco’s modified Eagle medium (DMEM, C11995500BT, Gibco, New York, NY, USA). The digestion reaction was terminated by 12% fetal bovine serum (FBS, 10099141, Gibco, New York, NY, USA). After centrifuge and filtration, Müller cells were cultured in 25 cm^2^ plastic culture flasks in a humidified condition of 95% air and 5% CO_2_ at 37 °C.

### 4.10. Cell Counting Kit-8 Assay

CCK-8 assays were performed to detect cell viability using the Cell Counting Kit-8 (BS350-B, Biosharp, Hefei, China). The primary RGCs and Müller cells were seeded onto 96-well plates for 24 h. Then, the required dose of drugs or H_2_O_2_ were added. After 24 h culture, 10 μL of CCK-8 reagent were incubated in each well at 37 °C for 1 h. The absorbance value was detected using the Filter Max F5 Microplate Reader (Molecular Devices, San Jose, CA, USA) at 450 nm wavelength.

### 4.11. Calcein-AM and Propidium Iodide (PI) Double Staining

PI/Calcein-AM double stain kit (C2015S, Beyotime, Shanghai, China) was used to detect cell apoptosis. After the required treatment, Müller cells or RGC were washed with PBS three times, followed by an additional 10 μM PI, 10 μM Calcein-AM, and 3 μM Hoechst 33342 (C1022, Beyotime, Shanghai, China) for each well at 37 °C for 15 min. Finally, these cells were washed twice with PBS and observed under a fluorescent microscope.

### 4.12. EdU Incorporation Assay

To detect cell proliferation, 5-Ethynyl-2’-deoxyuridine (EdU) assays were carried out with the BeyoClickTMEdU-555 detection kit (C0075S, Beyotime, Shanghai, China). The cells were incubated with 50 mM of EdU medium at 37 °C for 2 h. Subsequently, they were fixed with 4% paraformaldehyde for 15 min, washed three times with PBS for 5 min, permeabilized with 0.3% Triton X-100 for 15 min, and incubated with the Click reaction cocktail for 30 min. The nuclei were stained with DAPI. The fluorescence images were captured under a fluorescence microscope. The total number of cells and the number of EdU-positive cells were counted using Image J software (v1.53c).

### 4.13. Rhodamine123 Staining

To detect the change in the membrane potentials by Rhodamine 123 staining, Müller cells were seeded onto 24 well plates and stained with Rhodamine (BL931A, Biosharp, Hefei, China) plus Hoechst 33342 (C0071S-6, Beyotime, Shanghai, China) for 30 min at 37 °C. The stained cells were observed by a fluorescence microscope, and the fluorescence intensity was quantified by Image J software (v1.53c).

### 4.14. Flow Cytometry Analysis

For flow cytometry analysis, the Apoptosis Detection kit (A211-01, Vazyme, Nanjing, China) was used to detect the apoptosis of Müller cells and RGCs. Briefly, 200 μL of binding buffer were used to re-suspend cell pellets. Then, 2.5 μL of Annexin-V-FITC and 2.5 μL of PI working solution were added for the reaction for 20 min at room temperature according to the manufacturer’s instruction. Flow cytometry analysis was conducted by a flow cytometer (Beckman Coulter, Indianapolis, IN, USA).

### 4.15. Western Blot

The samples were collected and lysed in the cold radioimmunoprecipitation assay (RIPA) protein lysis buffer (P0013K, Beyotime, Shanghai, China) containing protease and phosphatase inhibitor cocktails (78440, Thermo Scientific, Waltham, MA, USA). Subsequently, the samples were centrifuged at 12,000 rpm at 4 °C for 10 min to obtain the supernatants. The protein concentration was determined by Micro BCA Protein Assay Kit (23227, Thermo Scientific, Waltham, MA, USA). The protein samples were boiled for 10 min, separated on 10% polyacrylamide SDS-PAGE gels, and transferred to polyvinylidene fluoride (PVDF) membranes (1620177, Bio-Rad, Hercules, CA, USA). The membranes were blocked with 5% non-fat milk in 0.1% Tween-20/Tris-buffered saline for 1 h at room temperature, incubated with the primary antibodies overnight at 4 °C, incubated with the HRP-conjugated secondary antibody (A0208, Beyotime, Shanghai, China) for 2 h at room temperature, and visualized by the enhanced chemiluminescence (ECL) Western kit (BI-WB004, sbjbio, Nanjing, China). The used antibodies are shown below: JNK (1:1000 dilution; ET1601-28, HuaBio, Hangzhou, China), phospho-JNK (1:1000 dilution; ET1609-42, HuaBio, Hangzhou, China), p38 (1:1000 dilution; ET1602-26, HuaBio, Hangzhou, China), phospho-p38 (1:1000 dilution; ET2001-52, HuaBio, Hangzhou, China), Bcl-2 (1:4000 dilution; ET1702-53, HuaBio, Hangzhou, China), Bax (1:2000 dilution; ET1603-34, HuaBio, Hangzhou, China), and GAPDH (1:1000 dilution; ET1601-4, HuaBio, Hangzhou, China).

### 4.16. Screening for the Targets of EYE-503

Network pharmacology was used to screen for the potential targets of EYE-503. The potential targets of EYE-503 were retrieved and predicted by the SUPERPRED database (https://prediction.charite.de/, accessed on 20 July 2020) and SwissTargetPrediction platform (http://www.swisstargetprediction.ch/) based on the chemical structure. The duplicate targets and non-human targets were removed. The protein names were converted to the gene names using the UniProt database (https://www.uniprot.org/, accessed on 20 July 2020). The DrugBank (https://www.drugbank.ca/, accessed on 20 July 2020), OMIM (https://www.omim.org/, accessed on 20 July 2020), and GeneCard (shttps://www.genecards.org/, accessed on 20 July 2020) databases were used to obtain the disease targets with the keyword of “Retinal neurodegeneration” after selecting human-derived targets, integrating, and removing duplicate targets. The predicted targets of EYE-503 were intersected to obtain the common drug-disease targets.

### 4.17. PPI Network Analysis

To identify the interactions among the different target proteins, the selected potential drug-disease targets were imported into the STRING 11.0 database (https://string-db.org/, accessed on 20 July 2020). The above-mentioned data were introduced into Cytoscape 3.6.1 for visualization. A PPI network diagram of the targets of EYE-503 was constructed.

### 4.18. GO Enrichment and KEGG Pathway Analysis

To understand the main action processes of the targets and the distribution in the pathway, GO enrichment (including biological process (BP), cellular component (CC), and molecular function (MF)) and KEGG pathway analysis were conducted using the metascape (https://metascape.org/, accessed on 20 July 2020) database. *p* < 0.05 indicated that the co-action targets had significant differences. The top 8 items of GO enrichment and the top 10 items of KEGG pathway analysis were mapped as bar plots and bubble plots by R version 4.0.2 software, respectively.

### 4.19. Statistical Analysis

All data were presented as mean ± standard deviation (SD) and analyzed using SPSS version 23 software and GraphPad Prism 8.0. Student’s t-test or one-way ANOVA followed by Bonferroni’s test was used to evaluate the statistical differences for the pair-wise or multiple comparisons. *p* < 0.05 was considered to be statistically significant.

## 5. Conclusions

This study confirms that EYE-503 administration contributes to the survival of RGCs and preserves visual function in the ONC model by blocking the JNK/p38 signaling pathway. This study provides a novel insight into the therapeutic strategy for retinal neurodegeneration.

## Figures and Tables

**Figure 1 pharmaceuticals-16-01033-f001:**
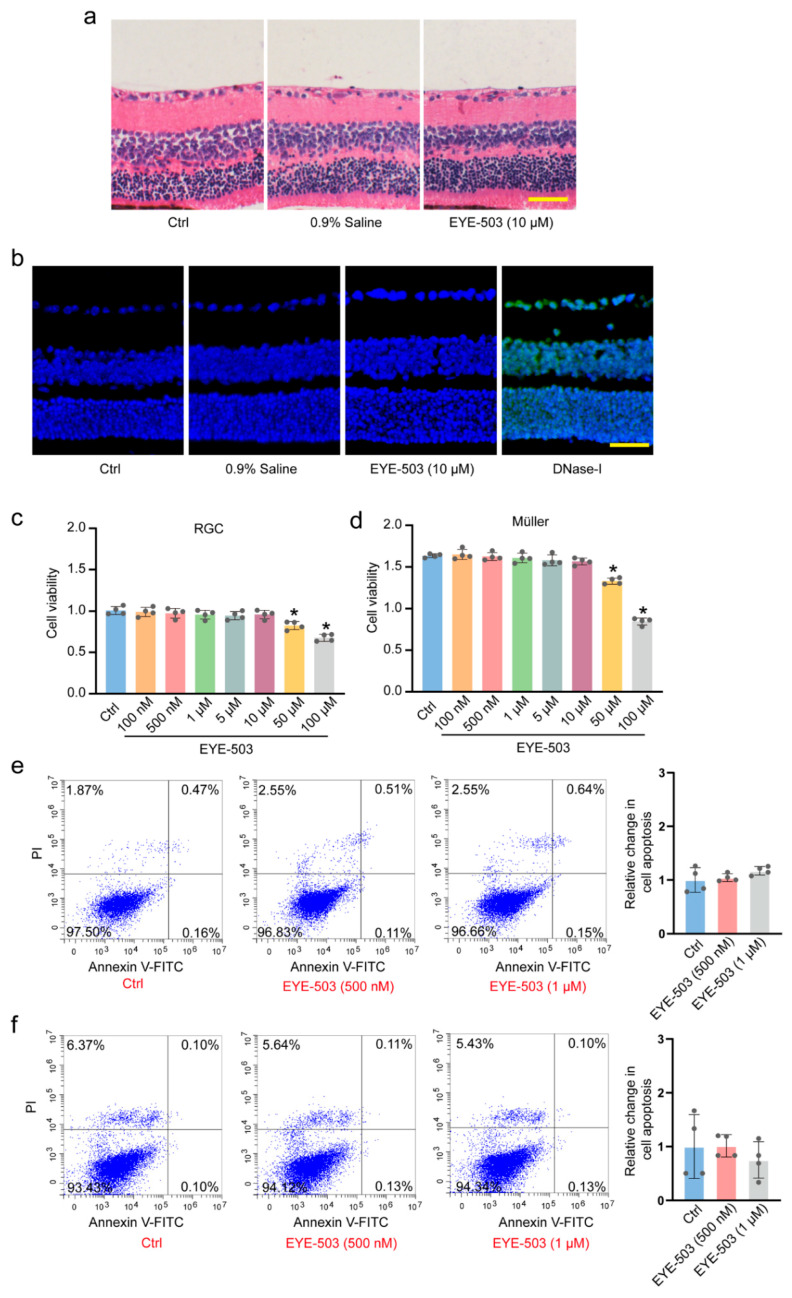
EYE-503 has no detectable cytotoxicity or tissue toxicity. (**a**,**b**) The retinas were treated with 0.9% saline, EYE-503 (10 μM), or left untreated (Ctrl) for 7 days. H&E staining was conducted to detect the change in retinal histological structures (**a**) (*n* = 6). TUNEL staining was performed to detect retinal apoptosis. Positive control group was treated with DNase-I reaction solution (C1082) for 15 min. Nuclei, blue; TUNEL, green (**b**) (n = 6). Scale bar: 50 μm. (**c**,**d**) RGCs and Müller cells were cultured with different concentrations of EYE-503 (100 nM–100 μM) or left untreated (Ctrl) for 24 h. Cell viability was detected by CCK-8 assays (*n* = 4; * *p* < 0.05 versus Ctrl; one-way ANOVA). (**e**,**f**) Annexin V-FITC/PI double-staining assays were performed to detect the apoptosis of RGCs and Müller cells (*n* = 4).

**Figure 2 pharmaceuticals-16-01033-f002:**
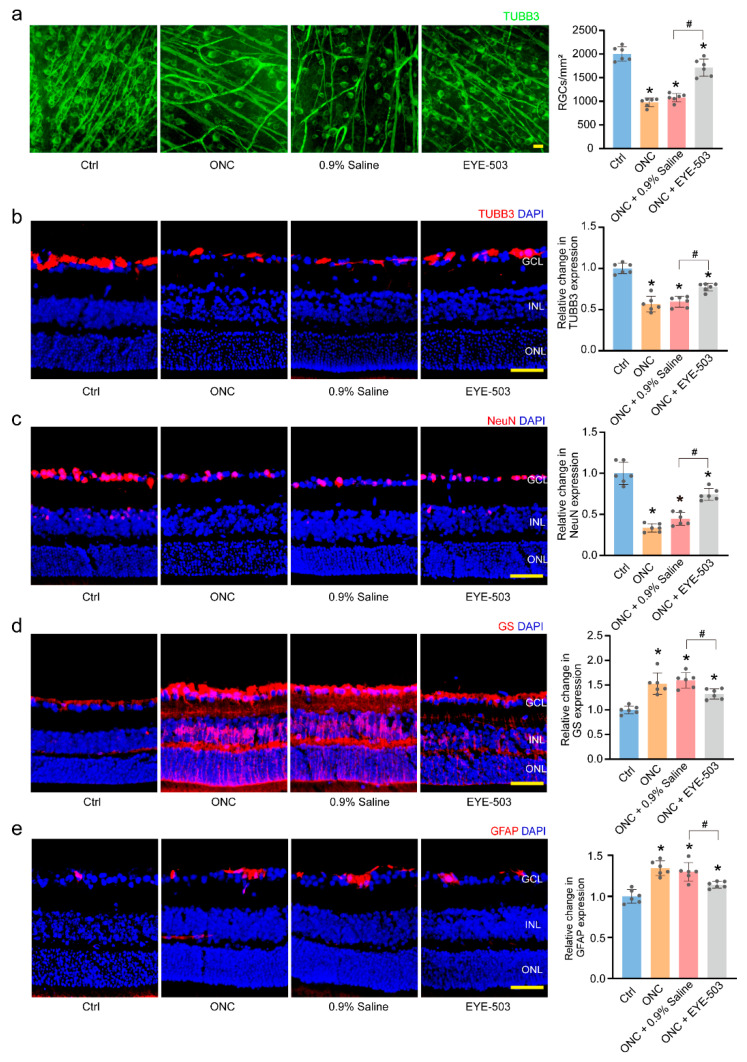
EYE-503 administration inhibits retinal reactive gliosis and contributes to RGC survival in vivo. C57BL/6J mice received an intravitreal injection of EYE-503 (10 μM), 0.9% saline (vehicle control), or were left untreated. Then, ONC models were established for 7 days. RGC survival was determined in the flat-mount retina by TUBB3 labeling (**a**) *n* = 6, Scale bar: 20 μm; * *p* < 0.05 versus Ctrl; ^#^
*p* < 0.05 ONC plus EYE-503 versus ONC plus 0.9% saline; one-way ANOVA). Immunofluorescence assays of TUBB3 (**b**), NeuN (**c**), GS (**d**), and GFAP (**e**) were conducted to detect RGC survival and retinal reactive gliosis. The representative images and quantitative analysis results were shown. Nuclei, blue; TUBB3, red; NeuN, red; GS, red; GFAP, red (*n* = 6, Scale bar: 50 μm; * *p* < 0.05 versus Ctrl; ^#^
*p* < 0.05 ONC plus EYE-503 versus ONC plus 0.9% saline; one-way ANOVA).

**Figure 3 pharmaceuticals-16-01033-f003:**
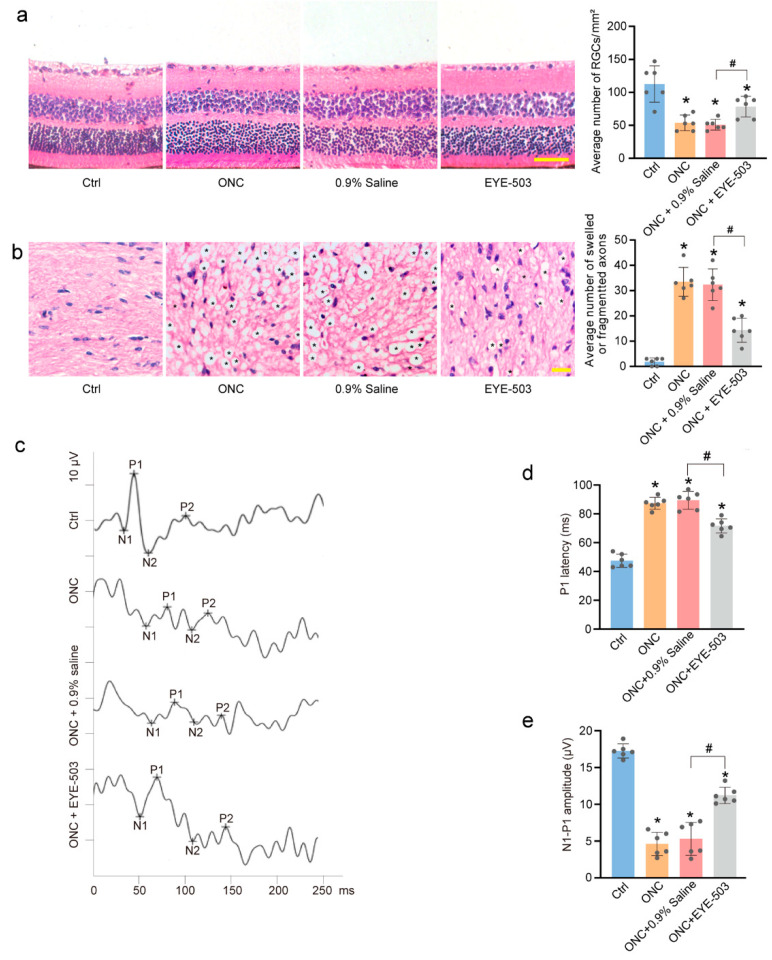
EYE-503 administration alleviates ONC-induced retinal injury and optic nerve injury in vivo. (**a**,**b**) C57BL/6J mice received an intravitreal injection of EYE-503 (10 μM), 0.9% saline (vehicle control) immediately following ONC or were left untreated as the control group. Seven days after ONC induction, H&E staining was performed to detect the number of retinal ganglion cells in the GCL layer (**a**); (*n* = 6, Scale bar: 50 μm; * *p* < 0.05 versus Ctrl; ^#^
*p* < 0.05 ONC plus EYE-503 versus ONC plus 0.9% saline; one-way ANOVA) and the degeneration of myelin sheath in the injured optic nerves (**b**);(*n* = 6, Scale bar: 20 μm; * *p* < 0.05 versus Ctrl; ^#^
*p* < 0.05 ONC plus EYE-503 versus ONC plus 0.9% saline; one-way ANOVA). (**c**) Representative waves of F-VEP in different groups. (**d**,**e**) The latency of P1 and the amplitude of N1-P1 were detected to reflect visual conduction function (*n* = 6; * *p* < 0.05 versus Ctrl; ^#^
*p* < 0.05 ONC plus EYE-503 versus ONC plus 0.9% saline; one-way ANOVA).

**Figure 4 pharmaceuticals-16-01033-f004:**
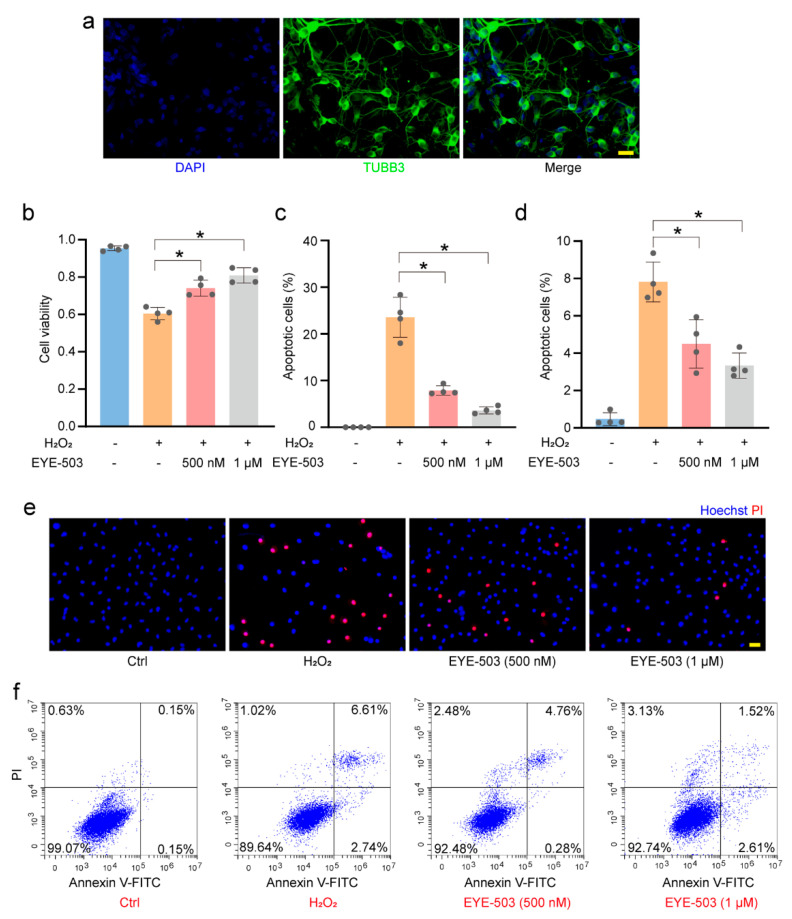
EYE-503 administration regulates RGC function in vitro. (**a**) Primary RGCs were identified with TUBB3 immunofluorescence (TUBB3, green; nuclei, blue. Scale bar: 20 μm). (**b**) Primary RGCs were exposed to H_2_O_2_ (200 μM), H_2_O_2_ plus EYE-503 (500 nM and 1 μM), or left untreated (Ctrl) for 24 h. CCK-8 assays were conducted to detect the viability of RGCs (*n* = 4; * *p* < 0.05 versus H_2_O_2_; one-way ANOVA). (**c**–**f**) The apoptosis of RGCs was determined using PI/Hoechst staining (**c**,**e**); *n* = 4; Scale bar: 20 μm; * *p* < 0.05 versus H_2_O_2_; one-way ANOVA) and annexin V-FITC/PI double labeling assays (**d**,**f**); (*n* = 4; * *p* < 0.05 versus H_2_O_2_; one-way ANOVA). PI, red; Hoechst, blue.

**Figure 5 pharmaceuticals-16-01033-f005:**
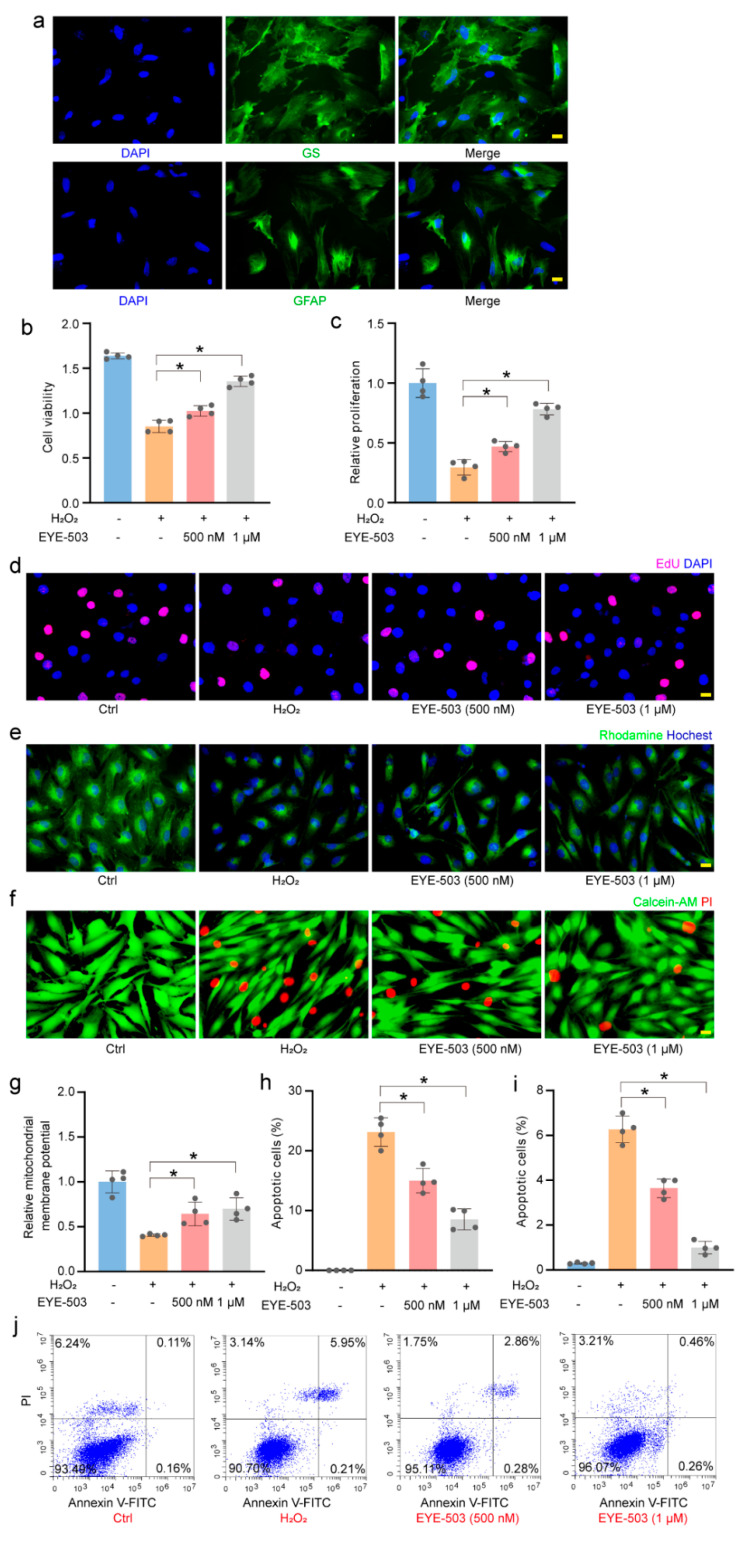
EYE-503 administration regulates Müller cell function in vitro. Primary Müller cells were exposed to H_2_O_2_ (200 μM), H_2_O_2_ plus EYE-503 (500 nM and 1 μM), or left untreated (Ctrl) for 24 h. Primary Müller cells were identified by GS and GFAP immunofluorescence staining (**a**) (GS, green; GFAP, green; nuclei, blue. Scale bar: 20 μm). CCK-8 assays were performed to detect the viability of Müller cells (**b**) (*n* = 4; * *p* < 0.05 versus H_2_O_2_; one-way ANOVA). EdU incorporation assays were performed to detect cell proliferation. Cell proliferation was quantified by calculating EdU-positive cells. EdU, red; nuclei, blue (**c**,**d**); (*n* = 4; Scale bar: 20 μm; * *p* < 0.05 versus H_2_O_2_; one-way ANOVA). Rhodamine staining was used to detect the change in mitochondrial membrane potentials. Rhodamine, green; nuclei, blue (**e**,**g**); (*n* = 4; Scale bar: 20 μm; * *p* < 0.05 versus H_2_O_2_; one-way ANOVA). The apoptosis of Müller cells was determined by PI/Calcein-AM staining (**f**,**h**); (*n* = 4; Scale bar: 20 μm; * *p* < 0.05 versus H_2_O_2_; one-way ANOVA) and annexin V-FITC/PI double labeling assays (**i**,**j**); (*n* = 4; * *p* < 0.05 versus H_2_O_2_; one-way ANOVA). PI, red; Calcein-AM, green.

**Figure 6 pharmaceuticals-16-01033-f006:**
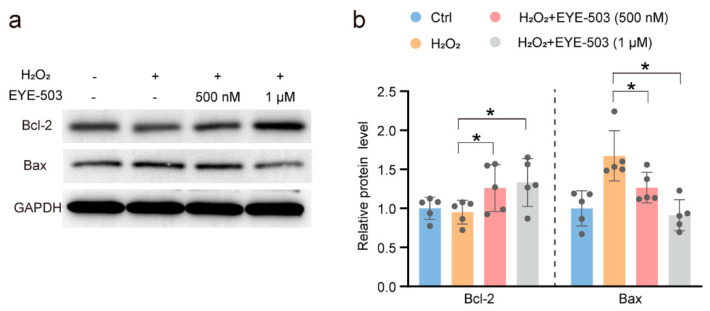
EYE-503 induced apoptotic-related protein Bcl-2 downregulation and Bax translocation. (**a**,**b**) The expression levels of Bcl-2 and Bax protein in retinal ganglion cells were detected by Western blots. GAPDH was detected as the internal control (*n* = 5, * *p* < 0.05 versus H_2_O_2_; one-way ANOVA).

**Figure 7 pharmaceuticals-16-01033-f007:**
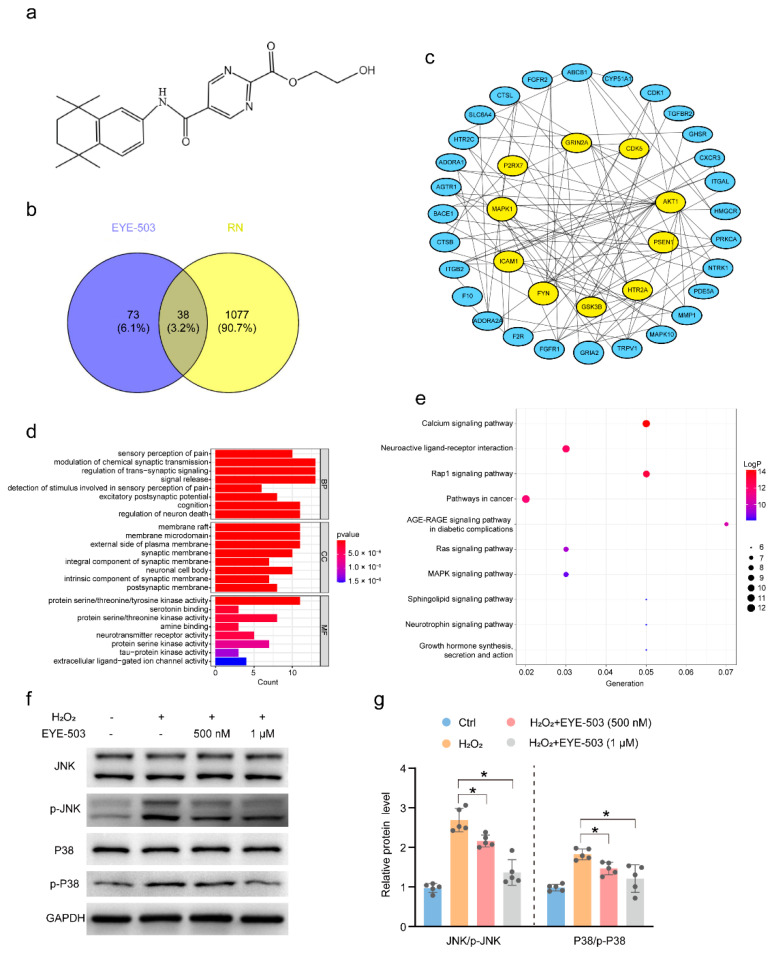
Potential targets of EYE-503 involved in retinal neurodegeneration. (**a**) The chemical structure of EYE-503. (**b**) Venn diagram showing the intersection target of EYE-503 and retinal neurodegeneration. (**c**) PPI network of the potential targets of EYE-503 during retinal neurodegeneration. (**d**) Bar plot diagram of GO enrichment analysis. (**e**) The top 10 items bubble plot of KEGG pathway analysis. (**f**) Western blot analysis of p38, p-p38, JNK, and p-JNK in retinal ganglion cells. GAPDH was used for normalization. (**g**) The phosphorylation levels of p38 and JNK in RGCs were evaluated by densitometric analysis. Relative p-p38 and p-JNK levels were calculated as the phosphorylated proteins levels divided by total proteins levels (*n* = 5, * *p* < 0.05 versus H_2_O_2_; one-way ANOVA).

## Data Availability

Data is contained within the article.

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
