# Peer review of "EYE-503: A Novel Retinoic Acid Drug for Treating Retinal Neurodegeneration"

_pharmaceuticals, 2023, doi:10.3390/ph16071033_

Round 1
Reviewer 1 Report
In this study, the authors assessed the potential of EYE-503, a novel drug for RA, to enhance the survival of retinal ganglion cells (RGCs) and preserve visual function. They conducted experiments using a mouse optic nerve crush model and in vitro models of RGC and Müller glial cells damage. The authors concludes that EYE-503 administration attenuated retinal ganglion cells apoptosis, inhibited reactive gliosis, and retarded the progression of retinal neurodegeneration, and that these findings provide valuable insights into therapeutic strategies for retinal neurodegeneration. While the manuscript is well-written, I have some issues that the authors could address.
1) Why did the authors use 10uM in vivo? Have the authors tried other concentrations?
2) On page 4, on the line 86 says “(a and b) The retinas were treated with 0.9% saline, EYE-503 (500 nM or 1 μM)”, but the concentration used was 10uM in vivo. Please correct it. In addition, in the lines 89-90 says “(c and d) RGCs and Müller cells were cultured with the tested concentrations of EYE-503 or left untreated (Ctrl) for 24 h”, please mention concentrations.
3) On page 4, on the lines 88-89 says that TUNEL staining was used. However, one micrograph mentions DNAse-1 without any indication in the legend or in the Materials and Methods section. It is essential to provide clear indications and clarify its usage as a positive control for TUNEL, in order to determine whether the compound does not induce apoptosis in any retinal cell type
4) On page 8, on the line 159, says “Primary mouse RGCs were cultured and identified with TUBB3 staining”, please include cites about TUBB3 as a RGC marker.
5) On page 15, on the lines 390-396, the authors provided a brief description of the process for obtaining GRC cultures. Please include cites.
6) On page 15, on the lines 395-396 says “The purifed RGCs were triturated with pipete and cultured in the complete medium (CM-122, Procell, Wuhan, China), however the MC-122 medium, Procell is not found on the web to see its composition. Please correct it or describe it.
7) GS is a marker that may or not be increased in retinopathies. It is advisable to describe if in the ONC is increased and add cites.
8) GFAP is a marker of glial reactivity. In figure 2 e, GFAP expression is observed in the GRC layer but not in the other. Why? Do the authors have other photos of other slices?
9) In the figure 7 g, please specify the culture type in which the study was conducted.
10) Does EYE-503 exhibit UV photosensitivity and cause significant non-targeted side effects? Is this the first study involving this drug? It would be beneficial to discuss in the manuscript the advantages of using this drug for RA, in addition to its water solubility, if that information is available."
11) RA exerts its molecular actions mainly through RAR and RXR nuclear receptors, do the authors know if EYE-503 acts as a ligand for that receptors? Please discuss in the manuscript.
Author Response
1) Why did the authors use 10 µM in vivo? Have the authors tried other concentrations?
Response:
Thanks for your suggestion.
The recommended concentration range of the drug is 0.1 μM to 10 μM according to manufacturers. In this study, we have established that concentrations up to 10 μM are safe for in vivo use. To ensure optimal therapeutic efficacy, we chose the concentration of 10 μM for validation. Thanks for your comments, we will continue to verify other concentrations in our subsequent studies.
2) On page 4, on the line 86 says “(a and b) The retinas were treated with 0.9% saline, EYE-503 (500 nM or 1 μM)”, but the concentration used was 10 μM in vivo. Please correct it. In addition, in the lines 89-90 says “(c and d) RGCs and Müller cells were cultured with the tested concentrations of EYE-503 or left untreated (Ctrl) for 24 h”, please mention concentrations.
Response:
Thanks for your suggestion.
On the line 85, we corrected the drug concentration to 10 μM. On the line 90, we added the drug concentration (100 nM - 100 μM).
3) On page 4, on the lines 88-89 says that TUNEL staining was used. However, one micrograph mentions DNAse-1 without any indication in the legend or in the Materials and Methods section. It is essential to provide clear indications and clarify its usage as a positive control for TUNEL, in order to determine whether the compound does not induce apoptosis in any retinal cell type.
Response:
Thanks for your suggestion.
According to the manufacturer's instructions, DNase-I shears genomic DNA to produces fragmentation. Cells with fragmented genomes exhibit positive staining when examined by TUNEL staining. It is widely used as a positive control for TUNEL staining in research [1-3].
We have added the description of DNase-I to the methods and legend section of the manuscript as follows:
Materials and Methods
4.6. Terminal deoxynucleotidyl transferase dUTP nick-end labeling (TUNEL) assay
Apoptosis was detected using the One Step TUNEL Apoptosis Assay Kit (C1088, Beyotime, Shanghai, China). Paraffin sections were de-paraffinized by xylene and etha-nol treatment, and then cultivated in proteinase K working solution (20 μg/mL) for 15 min at room temperature. Positive control group was treated with DNase-I reaction solution (C1082) for 15 min. Subsequently, the sections were washed with PBS and incubated with TUNEL reaction mixture at 37℃ for 1 h. The nuclear morphology of TUNEL-positive cells was stained with 4, 6-diamidino-2-phenylindole (DAPI, 1:1500 dilution, C1002, Biosharp, Hefei, China). The images were captured using a fluorescence microscope (IX73-TH4-200, Olympus, Japan).
Legend
Figure 1. EYE-503 has no detectable cytotoxicity and tissue toxicity.
(a and b) The retinas were treated with 0.9% saline, EYE-503 (10 μM), or left untreated (Ctrl) for 7 days. H&E staining was conducted to detect the changes of retinal histological structures (a, n = 6). TUNEL staining was used to detect retinal cell apoptosis. DNase-I treated groups were used as positive controls. Nuclei, blue; TUNEL, green (b, n = 6). Scale bar: 50 μm.
References:
- Liu C, Zhang CW, Zhou Y, Wong WQ, Lee LC, Ong WY, et al. APP upregulation contributes to retinal ganglion cell degeneration via JNK3. Cell Death Differ. 2018, 25, 663-678.
- Georgiadou E, Muralidharan C, Martinez M, Chabosseau P, Akalestou E, Tomas A, et al. Mitofusins Mfn1 and Mfn2 Are Required to Preserve Glucose- but Not Incretin-Stimulated β-Cell Connectivity and Insulin Secretion. Diabetes. 2022, 71, 1472-1489.
- Xiong X, Zhang L, Fan M, Han L, Wu Q, Liu S, et al. β-Endorphin Induction by Psychological Stress Promotes Leydig Cell Apoptosis through p38 MAPK Pathway in Male Rats. Cells. 2019, 8.
4) On page 8, on the line 159, says “Primary mouse RGCs were cultured and identified with TUBB3 staining”, please include cites about TUBB3 as a RGC marker.
Response:
Thanks for your suggestion.
The references we included in the manuscript are as follows. We have cited these two references on the line 160.
References:
- Navneet S, Zhao J, Wang J, Mysona B, Barwick S, Ammal Kaidery N, et al. Hyperhomocysteinemia-induced death of retinal ganglion cells: The role of Muller glial cells and NRF2. Redox Biol. 2019, 24, 101199.
- Rosignol I, Villarejo-Zori B, Teresak P, Sierra-Filardi E, Pereiro X, Rodriguez-Muela N, et al. The mito-QC Reporter for Quantitative Mitophagy Assessment in Primary Retinal Ganglion Cells and Experimental Glaucoma Models. Int J Mol Sci. 2020, 21.
5) On page 15, on the lines 390-396, the authors provided a brief description of the process for obtaining RGC cultures. Please include cites.
Response:
Thanks for your suggestion.
The references concerning the process for obtaining RGC cultures are as follows. We have cited these two references on the line 405.
References:
- Rosignol I, Villarejo-Zori B, Teresak P, Sierra-Filardi E, Pereiro X, Rodriguez-Muela N, et al. The mito-QC Reporter for Quantitative Mitophagy Assessment in Primary Retinal Ganglion Cells and Experimental Glaucoma Models. Int J Mol Sci. 2020, 21.
- Dun Y, Mysona B, Van Ells T, Amarnath L, Ola MS, Ganapathy V, et al. Expression of the cystine-glutamate exchanger (xc-) in retinal ganglion cells and regulation by nitric oxide and oxidative stress. Cell Tissue Res. 2006, 324, 189-202.
6) On page 15, on the lines 395-396 says “The purified RGCs were triturated with pipette and cultured in the complete medium (CM-122, Procell, Wuhan, China), however the MC-122 medium, Procell is not found on the web to see its composition. Please correct it or describe it.
Response:
Thanks for your suggestion.
We apologize for writing the wrong medium used. The medium we used was CM-M122, which can be found on the website (https://www.procell.com.cn). We have corrected the medium on the line 405 in the revised manuscript.
7) GS is a marker that may or not be increased in retinopathies. It is advisable to describe if in the ONC is increased and add cites.
Response:
Thanks for your suggestion.
The key features of retinal neurodegeneration are RGC apoptosis and reactive gliosis [1]. Although it includes a variety of diseases, the pathological process of neurodegeneration is similar [2]. Optic nerve crush is a common model in retinal neurodegeneration. It has been found that the expression of GS is increased in retinal neurodegeneration [3-5], which is consistent with our findings.
We have cited the corresponding references on the lines 105-106.
References:
- Cuenca N, Fernández-Sánchez L, Campello L, Maneu V, De la Villa P, Lax P, et al. Cellular responses following retinal injuries and therapeutic approaches for neurodegenerative diseases. Prog Retin Eye Res. 2014, 43, 17-75.
- Manogaran P, Samardzija M, Schad AN, Wicki CA, Walker-Egger C, Rudin M, et al. Retinal pathology in experimental optic neuritis is characterized by retrograde degeneration and gliosis. Acta Neuropathol Commun. 2019, 7, 116.
- Bringmann A, Grosche A, Pannicke T, Reichenbach A. GABA and Glutamate Uptake and Metabolism in Retinal Glial (Müller) Cells. Front Endocrinol (Lausanne). 2013, 4, 48.
- Toops KA, Hagemann TL, Messing A, Nickells RW. The effect of glial fibrillary acidic protein expression on neurite outgrowth from retinal explants in a permissive environment. BMC Res Notes. 2012, 5, 693.
- Reinehr S, Koch D, Weiss M, Froemel F, Voss C, Dick HB, et al. Loss of retinal ganglion cells in a new genetic mouse model for primary open-angle glaucoma. J Cell Mol Med. 2019, 23, 5497-5507.
8) GFAP is a marker of glial reactivity. In figure 2 e, GFAP expression is observed in the GRC layer but not in the other. Why? Do the authors have other photos of other slices?
Response:
Thanks for your suggestion.
Glial cells are reactively activated during retinal neurodegeneration, leading to retinal cell damage. Activated glial cells are highly accumulated at the site of injury and increase the expression of GFAP. GFAP is used as a marker of glial cell activation and reflects the extent of activation. Optic nerve crush is known to induce glial cell accumulation at the injury site, leading to reactive gliosis at the injured sites [1]. In fact, a well-known feature of astrocyte activation and reactive gliosis are the increased production of intermediate filament proteins and remodeling of the intermediate filament system of astrocytes. Activation of astrocytes is associated with the changes in the expression of many genes and characteristic morphological hallmarks and has important functional consequences in situations such as trauma. Reactive gliosis has been described as the constitutive, graded, multi-stage, and evolutionary conserved defensive astroglial reaction [2-4].
In our study, we used an optic nerve crush model to investigate whether EYE-503 administration affected the expression of GFAP. Following optic nerve crush, GFAP immunofluorescent staining was performed on retinal sections. As shown in Figure 2e, ONC trauma led to increased expression of GFAP, while EYE-503 administration could downregulate the induction of GFAP. Moreover, increased GFAP was detected in GCL layer.
In the previous studies about neurodegeneration, GFAP was mainly expressed in the GCL of retinal sections, which is consistent with the results in this study. We also conducted literature search to detect the expression pattern of GFAP. As shown in the following references, increased GFAP was mainly detected in GCL layer.
- Zhang J, Liu Z, Wu H, et al. Irisin attenuates pathological neovascularization in oxygen-induced retinopathy mice. Invest Ophthalmol Vis Sci. 2022; 63(6):21.
(2) Chen et al. Interphotoreceptor retinol-binding protein ameliorates diabetes-induced retinal dysfunction and neurodegeneration through Rhodopsin. Diabetes. 2021; 70(3):788-799.
References:
- Lye-Barthel M, Sun D, Jakobs TC. Morphology of astrocytes in a glaucomatous optic nerve. Invest Ophthalmol Vis Sci. 2013, 54, 909-917.
- Lin SF, Chien JY, Kapupara K, Huang CF, Huang SP. Oroxylin A promotes retinal ganglion cell survival in a rat optic nerve crush model. PLoS One. 2017, 12, e0178584.
- Chen J, Shao Y, Sasore T, Moiseyev G, Zhou K, Ma X, et al. Interphotoreceptor Retinol-Binding Protein Ameliorates Diabetes-Induced Retinal Dysfunction and Neurodegeneration Through Rhodopsin. Diabetes. 2021, 70, 788-799.
- Zhang J, Liu Z, Wu H, Chen X, Hu Q, Li X, et al. Irisin Attenuates Pathological Neovascularization in Oxygen-Induced Retinopathy Mice. Invest Ophthalmol Vis Sci. 2022, 63, 21.
9) In the figure 7 g, please specify the culture type in which the study was conducted.
Response:
Thanks for your suggestion.
This experiment was performed using RGCs, and we examined the total protein and phosphorylated protein levels in the cells.
The relevant changes are highlighted in red in the revised manuscript on the lines 246 and 257.
10) Does EYE-503 exhibit UV photosensitivity and cause significant non-targeted side effects? Is this the first study involving this drug? It would be beneficial to discuss in the manuscript the advantages of using this drug for RA, in addition to its water solubility, if that information is available."
Response:
Thanks for your suggestion.
In this study, we focused on studying the therapeutic effects of EYE-503 on retinal neurodegeneration. Our primary study has revealed that EYE-503 has no detectable cytotoxicity and tissue toxicity as shown in Figure 1. We have shown that EYE-503 can promote RGC survival and inhibit reactive gliosis, thereby offering a potential therapeutic approach for treating retinal neurodegeneration. We appreciate your valuable comments. We totally agree with your suggestion that physicochemical properties such as UV photosensitivity and non-targeted side effects are still required to be investigated in our future studies.
Thanks for your suggestion again.
11) RA exerts its molecular actions mainly through RAR and RXR nuclear receptors, do the authors know if EYE-503 acts as a ligand for that receptors? Please discuss in the manuscript.
Response:
Thanks for your suggestion.
Based on the chemical structure of EYE-503 (Fig. 7a), the SwissTargetPrediction and Superpred database was used to search for the potential targets of EYE-503. After filtering out these duplicate targets, 111 potential targets of EYE-503 were retrieved. 1115 targets of retina neurodegeneration were obtained from the DrugBank, OMIM, and GeneCard databases. After mapping EYE-503-related targets and disease-related targets in a Venn diagram, the 38 overlapping targets were accessed (Fig. 7b). To identify the core proteins interaction of EYE-503 intervention for retinal neurodegeneration, 38 drug-disease common targets were inducted into the STRING database. The target network was made up of 38 nodes and 109 edges, with an average node degree of 5.74. PPI network diagram of EYE-503 against retinal neurodegeneration was constructed by Cytoscape 3.0.1 (Fig. 7c).
Next, the metascape database was used for KEGG pathway enrichment analysis to investigate the potential signaling pathways based on the 38 drug-disease common targets. The 10 most important KEGG signaling pathways related to retinal neurodegeneration with the condition of P < 0.05 were shown in Figure 7e. Calcium signaling pathway, Neuroactive ligand-receptor interaction, Rap1 signaling pathway, cancer pathway, MAPK signaling pathway, and neuroreophin signaling pathway were included. EYE-503 may act on these pathways against retinal neurodegeneration. Given MAPK signaling pathway can mediate several biological processes, such as proliferation, differentiation, senescence, and apoptosis, which are tightly associated with retinal neurodegeneration. We speculated that EYE-503 may act on MAPK signaling pathway against retinal neurodegeneration by network pharmacology. As shown in Figure 7f-g, stress stimulation led to an obvious increase in the phosphorylated levels of p38 and JNK in RGCs. We found that EYE-503 administration decreased the levels of phospho-JNK and phospho-p38 in various degrees. Collectively, we speculate that EYE-503 administration alleviates retinal cell apoptosis by suppressing JNK/p38 signaling.
We totally agree with your idea that RA exerts its molecular actions mainly through RAR and RXR nuclear receptors. EYE-503 is a retinoid compound. In our mechanistic studies, we utilized network pharmacology to identify potential signaling pathways linking EYE-503 and retinal neurodegeneration. However, among those signaling pathways, RARs/RXRs nuclear receptors are not the key signaling involved in retinal neurodegeneration. We thus focused on investigating the JNK/p38 signaling pathway, which exhibited greater significance than RARs/RXRs in our study. We found that EYE-503 can inhibit the level of phosphorylation of the JNK/p38 signaling pathway, thereby is a potential treatment for retinal neurodegeneration. Your valuable suggestion will be of great help to our research. We acknowledge the significance of studying EYE-503 as a nuclear receptor ligand and will include further investigations on this aspect in our future research.

Reviewer 2 Report
The study explored the effect of a novel retinoic acid drug called EYE-503 on optic crush model in vivo, RGC and Muller cells in vitro. Generally, it is well-designed study. The experiment on Muller cell in vitro is interesting. However, there are several concerns that should be addressed to improve the manuscript:
1 Line34-36, the definition of retinal degeneration is not accurate. It usually includes the loss of photoreceptors. The loss of RGC is called retinal neurodegeneration. This study demonstrated that EYE-503 contributes to the survival of RGCs in optic nerve crush model. It is better to clarify in the Abstract, as well.
2. What is the relationship or interaction between Muller cells and RGCs? Only the western blot results of RGCS in vitro in presented, how about the Muller cells? Since the two-part cell experiment was designed, it should be explored, and at least discussed in detail in the Discussion.
3. The primary culture of RGC from mouse is challenging. It often requires two RGC related markers to demonstrate the purity. Please explain why only TUBB3 staining was used.
4. The retinal flat mount counting of RGCs is described several times throughout the Results. However, only images of retinal sections are presented in the Figures. Please clarify which methods were used to count the RGCs.
5. GFAP, is considered to be an early and sensitive marker of retinal injury. Please explain why the GFAP expression is not marked upregulated after the optic nerve crush. (Figure 2e)
English expression is fine.
Author Response
Reviewer 2
The study explored the effect of a novel retinoic acid drug called EYE-503 on optic crush model in vivo, RGC and Muller cells in vitro. Generally, it is well-designed study. The experiment on Muller cell in vitro is interesting. However, there are several concerns that should be addressed to improve the manuscript:
- Line34-36, the definition of retinal degeneration is not accurate. It usually includes the loss of photoreceptors. The loss of RGC is called retinal neurodegeneration. This study demonstrated that EYE-503 contributes to the survival of RGCs in optic nerve crush model. It is better to clarify in the Abstract, as well.
Response:
Thanks for your suggestion.
In this study, we studied the mechanism of retinal neurodegeneration. We have defined it in the Abstract and introduction. The main characteristics of retinal neurodegeneration involve RGC apoptosis, axonal degeneration, and reactive gliosis.
Retinal neurodegenerative diseases like age-related macular degeneration, glaucoma, diabetic retinopathy or retinitis pigmentosa are the most frequent causes of incurable low vision and blindness worldwide. Retinal neurodegeneration may be caused by genetic defects, increased intraocular pressure, high levels of blood glucose or other types of stress or aging. All of them cause progressive neuronal death and activation of glial cells. Although the etiology, pathogenesis, and clinical characteristics of retinal neurodegenerative diseases are very different, they have common features because the cellular and molecular response to retinal neurodegeneration is closely similar. Thus, it had been proposed that several neuroprotective therapeutic approaches may be adequate for retinal neurodegenerative process.
- What is the relationship or interaction between Muller cells and RGCs? Only the western blot results of RGCs in vitro in presented, how about the Muller cells? Since the two-part cell experiment was designed, it should be explored, and at least discussed in detail in the Discussion.
Response:
Thanks for your suggestion. We have discussed this point in the discussion section.
The relationship between RGCs and Müller cells is highly significant. Structurally, the Müller cells extend upwards towards the inner limiting membrane to form a terminal peduncle that wraps around the dendrites and axons of RGCs. Physiologically, Müller cells release neurotrophic factors, gliotransmitters, and antioxidant factors to maintain metabolic coupling with RGCs. When the RGC is stressed, Müller cells undergo a complex cascade of structural and molecular changes that cause reactive gliosis. During retinal neurodegeneration, the irreversible loss of RGCs directly results in visual impairment. Therefore, our study focused on investigating the molecular mechanism of RGCs.
Müller cells are the major types of retinal glial cells, which are responsible for maintaining retinal homeostasis and ensure proper function of healthy retina. At the early stage of neurodegeneration, activated glial cells play a neuroprotective role in the retina by increasing the expression of cytoprotective factors or restoring neurotransmitter and ion balance. At the late stage, proliferative gliosis can accelerate retinal neurodegeneration, causing direct and indirect damages to the neurons and vasculature. These evidence suggests that reactive Müller cells may exert cytoprotective and cytotoxic effects on retinal neurons. Given Müller cells play a dual role in retinal neurodegeneration, we speculate that EYE-503 administration might affect the activation of Müller cells through different signaling pathways at different stages of retinal neurodegeneration. Because RGC loss was observed throughout the process of retinal neurodegeneration, we mainly investigated the mechanism of EYE-503-mediated RGC protection
References:
Li X, Liu J, Hoh J, Liu J. Müller cells in pathological retinal angiogenesis. Transl Res. 2019, 207, 96-106.
Vecino E, Rodriguez FD, Ruzafa N, Pereiro X, Sharma SC. Glia-neuron interactions in the mammalian retina. Prog Retin Eye Res. 2016, 51, 1-40.
- The primary culture of RGC from mouse is challenging. It often requires two RGC related markers to demonstrate the purity. Please explain why only TUBB3 staining was used.
Response:
Thanks for your suggestion.
As your suggestion, several markers have been used to label RGCs, such as MAP-2, TUBB3, RBPMS and THY1. MAP-2 encodes a protein that belongs to the microtubule-associated protein family. The proteins of this family are thought to be involved in microtubule assembly, which is an essential step in neurogenesis. THY1 gene encodes a cell surface glycoprotein and member of the immunoglobulin superfamily of proteins. The encoded protein is involved in cell adhesion and cell communication in numerous cell types, but particularly in cells of the immune and nervous system. The pan-RGC marker RBPMS is also expressed in a subset of microglia. Given TUBB3 is known as a universal marker of RGCs, we thus used TUBB3 staining for the identification of RGCs [1]. In fact, there are several studies that have solely used TUBB3 as a marker for the identification of primary RGCs [2, 3]. We appreciate your valuable comments. We also attempted to use MAP-2 to label RGCs and the purity of primary mouse RGCs was above 85%.
References:
1.Navneet S, Zhao J, Wang J, Mysona B, Barwick S, Ammal Kaidery N, et al. Hyperhomocysteinemia-induced death of retinal ganglion cells: The role of Muller glial cells and NRF2. Redox Biol. 2019, 24, 101199.
- Rosignol I, Villarejo-Zori B, Teresak P, Sierra-Filardi E, Pereiro X, Rodriguez-Muela N, et al. The mito-QC Reporter for Quantitative Mitophagy Assessment in Primary Retinal Ganglion Cells and Experimental Glaucoma Models. Int J Mol Sci. 2020, 21.
- Shi Y, Ye D, Huang R, Xu Y, Lu P, Chen H, et al. Down Syndrome Critical Region 1 Reduces Oxidative Stress-Induced Retinal Ganglion Cells Apoptosis via CREB-Bcl-2 Pathway. Invest Ophthalmol Vis Sci. 2020, 61, 23.
- The retinal flat mount counting of RGCs is described several times throughout the Results. However, only images of retinal sections are presented in the Figures. Please clarify which methods were used to count the RGCs.
Response:
Thanks for your suggestion.
In the retinal flat mount counting method, we divided the retina into four quadrants with the optic papilla as the center. Two images were taken from each quadrant at the same distance from the optic papilla. The total number of RGCs for each quadrant image was counted using Image J and averaged. The results are expressed as mm2.The relevant changes are highlighted in red in the revised manuscript.
- GFAP, is considered to be an early and sensitive marker of retinal injury. Please explain why the GFAP expression is not marked upregulated after the optic nerve crush. (Figure 2e)
Response:
Thanks for your suggestion.
Optic nerve crush is known to induce glial cell accumulation at the injury site, leading to reactive gliosis at the injured sites (Lye-Barthel et al., 2013, IOVS). In fact, a well-known feature of astrocyte activation and reactive gliosis are the increased production of intermediate filament proteins and remodeling of the intermediate filament system of astrocytes. Activation of astrocytes is associated with changes in the expression of many genes and characteristic morphological hallmarks, and has important functional consequences in situations such as trauma. Reactive gliosis has been described as constitutive, graded, multi-stage, and evolutionary conserved defensive astroglial reaction.
In our study, we used an optic nerve crush model to investigate whether EYE-503 administration affected the expression of GFAP. Following optic nerve crush, GFAP immunofluorescent staining was performed on retinal sections. As shown in Figure 2e, ONC trauma led to increased expression of GFAP, while EYE-503 administration could downregulate the induction of GFAP. Moreover, increased GFAP was detected in GCL layer.
We also conducted literature search to detect the expression pattern of GFAP. As shown in the following references, increased GFAP was also mainly detected in GCL layer.
- Zhang J, Liu Z, Wu H, et al. Irisin attenuates pathological neovascularization in oxygen-induced retinopathy mice. Invest Ophthalmol Vis Sci. 2022; 63(6):21.
(2) Chen et al. Interphotoreceptor retinol-binding protein ameliorates diabetes-induced retinal dysfunction and neurodegeneration through Rhodopsin. Diabetes. 2021; 70(3):788-799.

Reviewer 3 Report
I think the manuscript could be published.
Author Response
Thank you for your affirmation of our research. We greatly appreciate your positive feedback, and we will continue to conduct further in-depth research in this area.